# You are caught stealing my winning lottery ticket! Making a lottery ticket claim its ownership

**Xuxi Chen[1*], Tianlong Chen[1*], Zhenyu Zhang[2], Zhangyang Wang[1]**
[1]University of Texas at Austin, [2]University of Science and Technology of China
{xxchen,tianlong.chen,atlaswang}@utexas.edu,zzy19969@mail.ustc.edu.cn

## Abstract

Despite tremendous success in many application scenarios, the training and inference costs of using deep learning are also rapidly increasing over time. The lottery ticket hypothesis (LTH) emerges as a promising framework to leverage a special sparse subnetwork (i.e., *winning ticket*) instead of a full model for both training and inference, that can lower both costs without sacrificing the performance. The main resource bottleneck of LTH is however the extraordinary cost to find the sparse mask of the winning ticket. That makes the found winning ticket become a valuable asset to the owners, highlighting the necessity of protecting its copyright. Our setting adds a new dimension to the recently soaring interest in protecting against the intellectual property (IP) infringement of deep models and verifying their ownerships, since they take owners' massive/unique resources to develop or train. While existing methods explored encrypted weights or predictions, we investigate a unique way to leverage sparse topological information to perform *lottery verification*, by developing several graph-based signatures that can be embedded as credentials. By further combining trigger set-based methods, our proposal can work in both white-box and black-box verification scenarios. Through extensive experiments, we demonstrate the effectiveness of lottery verification in diverse models (ResNet-20, ResNet-18, ResNet-50) on CIFAR-10 and CIFAR-100. Specifically, our verification is shown to be robust to removal attacks such as model fine-tuning and pruning, as well as several ambiguity attacks. Our codes are available at https://github.com/VITA-Group/NO-stealing-LTH.

## 1 Introduction

Deep neural networks (DNNs) have dramatically raised the state-of-the-art performance in various fields. However, the over-parameterization of DNNs becomes a non-negligible problem. The amount of parameters now is often on the billion scale, which significantly increases the inference cost when using these models. An emerging field of *lottery ticket hypothesis* (LTH) explores a new scheme for pruning the model without sacrificing performance. The core idea is to identify the sparsity pattern ahead of training (or in its early stage) and train a sparse network from scratch. It has been hypothesized [1] that DNNs contain sparse networks named *winning tickets* that can be trained to match the test accuracy of the full model. These winning tickets hence have comparable or even better inference performance while potentially reducing the computational footprints.

However, finding winning tickets is a non-trivial task: it involves the training-prune-retraining cycle for several times [1], which is burdensome and computation-consuming. Although other works [2–4] have shown that sparsity might emerge at the initialization or at the early stage of training, the iterative magnitude pruning (IMP) still outperforms these alternatives by clear margins [5]. Yet, the powerful IMP method requires multiple rounds of train-prune-train process on the original training set, which

---

*Equal Contribution.

35th Conference on Neural Information Processing Systems (NeurIPS 2021).

is even much more expensive than training a dense network. That makes a found winning ticket a valuable asset to the owners, highlighting the necessity of protecting the winning tickets' copyright.

Previous works [6–9] have shown that deep networks are vulnerable to intellectual property (IP) infringement. For example, one can use transfer learning to adapt a trained model onto a new task or use model compression techniques to create a new sparse model based on the target model. Fortunately, in recent years the ownership verification problem has been addressed with a number of solutions proposed. The key idea is to embed verifiable information, or called *signatures*, into models' weights [6, 10, 11] or predictions [7] without visibly affecting the original performance. By extracting the embedded information from models, one can verify the ownership of models and hence protect their IPs. For the methods that embed information in weights, additional weights regularizers are often used to enforce certain patterns, such as signs. As for the prediction methods, a special training set, which is often called a *trigger* set, is used as additional training data. The model trained upon both the original data and the trigger set can generate desired prediction labels for the privately-held trigger set, while preserving the performance on the original training set. However, those general methods did not take any structural property(e.g., sparsity) into account, leaving chance for improving their gains in the winning ticket mask protection.

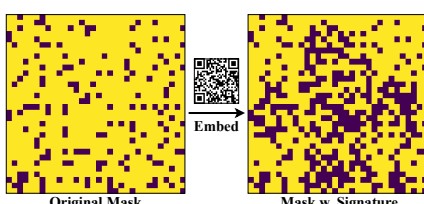

Figure 1: Illustration of embedding signatures into the original sparse mask. These visualizations are projected from 4D tensors. Dark entries are pruned elements. Note that the actual sparsity of the subnetwork is **unchanged** after encoding credentials.

On protecting the IP of winning tickets, we investigate a novel way to leverage **sparse structural information** for ownership verification (Fig. 1). This structural information embedded in winning tickets is a good "credential" for ownership verification since the winning ticket at extreme sparsity is naturally robust to fine-tuning and (further) pruning attacks. The winning ticket at extreme sparsity cannot be pruned further; otherwise, the inference performance will drop (hence losing its "value"). Meanwhile, fine-tuning the winning tickets can only tune the weights, but the sparsity pattern will not be changed. However, there remain some key questions to answer: *How to formulate the ownership verification process under the context of the lottery ticket hypothesis? What kind of structural information should be used? How to inject user-specific information into the structure of winning tickets?* We present answers to these questions in this paper. We summarize our findings as follows:

- We formulate the lottery verification problem and define two different protection scenarios. We show that even without specific protection, the extremely sparse winning ticket can partly claim its ownership because of the critical role of its sparse structure in the final inference performance.

- We further propose a new mask embedding method that is capable of embedding ownership "signatures" in the subnetwork's sparse structural connectivity (Fig. 1), without much affecting its performance. The signature is robust, e.g., it can be extracted and decoded even after pruning or fine-tuning attacks. Combined further with the trigger set-based method, our mask embedding method can work under both white-box and black-box verification frameworks.

- We investigate several verification schemes, *i.e.,* separate masks, embedding signatures, and embedding signatures with the trigger set. We show that these schemes are robust to the common removal and ambiguity attacks, as well as a new type of "add-on" attacks. Extensive experiment results demonstrate their competence on protecting the winning tickets. For example, on ResNet-20, our verification framework can defend fine-tuning attacks intrinsically, as well as pruning attacks and as add-on attack under all levels of pruning ratios.

## 2  Related Work

**Pruning and Lottery Ticket Hypothesis.**  Pruning algorithms can be roughly classified into two types: pruning *after* training and pruning *before* training. Conventional pruning after algorithms assign scores to individual weights and remove weights with the lowest scores [12]. There are a large number of scoring algorithms of this kind, including weights magnitude [13, 14], Taylor coefficients [15–18], and other variants [19, 20]. Pruning-before-training methods also play important roles in the community [1]. However, it requires multiple cycles of training and pruning process [12], making it sometimes tedious in practical use. Pruning-at-initialization methods alleviate such cost by

pruning initial weights such as observing initial gradients of weights [2, 3], but the quality of sparse networks found by these methods are mediocre in return.

A representative pruning-before-training method is the lottery ticket hypothesis [1], which hypothesizes the existence of a sparse network in dense networks that can be trained to match the test accuracy of dense networks in isolation. It was later verified that the original hypothesis was too strong, and early rewind techniques [21] help scale up the original version. Later on, the lottery ticket hypothesis and its variants have been explored and extended to various fields [22–30] such as image generation [31, 27, 32] and natural language processing [22, 24]. However, it is currently non-trivial to find winning tickets, especially at high sparsity since multiple train-prune-train processes are required, which suggests the practical value of protecting the found sparse networks.

**Ownership Verification.** Ownership verification has drawn attention from both the industry and academia. Many works have been proposed to address IP protection. One most popular way is to use watermarking algorithm: [6] proposed to embed watermarks in the form of bits into deep networks' weights by an additional regularization term. [10] embedded information into model weights by regularizing on the signs of weights. Besides watermarking on weights, [7, 33] embedded watermarks in the labels of certain examples in a *trigger set*, which makes it possible to extract watermarks through a service interface without directly accessing the models' weights (black-box setting). Following somehow different pathways, [34] proposed a passport-based approach that encodes signatures with special passport layers. [11] presented passport-embedded normalization whose parameters are associated with signatures. However, none of these methods leverages structural information for ownership verification besides assuming general dense networks. For sparse networks, the sparse mask patterns represent the key information and can vary across models and tasks.

## 3 The Lottery Ticket Claims its Ownership

### 3.1 The Lottery Ticket Hypothesis

❶ **Sparse Masks, Subnetworks and Winning Tickets.** We define a neural network parameterized by $\mathbf{W}$ as $\mathbb{N}[\mathbf{W}](\cdot)$. With a slight abuse of notion, we define a subnetwork of $\mathbb{N}[\mathbf{W}]$ as $\mathbb{N}[\mathbf{W}, \mathbf{M}] := \mathbb{N}[\mathbf{W} \odot \mathbf{M}](\cdot)$ where $\mathbf{M}$ is a sparse mask whose shape is the same as $\mathbf{W}$ but the value of each entry in which can only be either 0 or 1. Given $\mathbf{W}_0$ the initialization of $\mathbb{N}$, if $\mathbb{N}[\mathbf{W}_0, \mathbf{M}]$ can be **re-trained** to *match* the test performance of the dense model training from $\mathbb{N}[\mathbf{W}_0]$, we call $\mathbb{N}[\mathbf{W}_0, \mathbf{M}]$ a **winning ticket**. The term re-train above is used to distinguish between the training process to find the winning ticket. The criterion for matching can be set as, e.g., no lower than 1% than dense models' performance. ❷ **Sparsity Comparison.** The sparsity of a sparse mask $\mathbf{M}$ can be defined as $\mathrm{spar}(\mathbf{M}) := \|\mathbf{M}\|_0 / \|\mathbf{M} + 1\|_0$ where $\| \cdot \|_0$ represents the non-zero values of the input matrix, and the relative sparsity can be defined as $\mathrm{rspar}(\mathbf{M}_1, \mathbf{M}_2) := \mathrm{spar}(\mathbf{M}_1)/(\mathrm{spar}(\mathbf{M}_2) + \epsilon)$. We call a sparse mask $\mathbf{M}_1$ is *sparser* than $\mathbf{M}$ if and only if $\|\mathbf{M}_1\|_0 < \|\mathbf{M}\|_0$. A *sub-mask* $\mathbf{M}_2$ is a mask that has the same shape as $\mathbf{M}$ satisfying that all the elements in $\mathbf{M} - \mathbf{M}_2$ are non-negative. ❸ **Extremely Sparse Condition.** Given a sparsity difference threshold $t$, we call $\mathbb{N}[\mathbf{W}_0, \mathbf{M}_e]$ an **extremely sparse winning ticket** (or referred to as an **extreme ticket** hereinafter for conciseness) if $\mathbb{N}[\mathbf{W}_0, \mathbf{M}_e]$ can match the performance of $\mathbb{N}[\mathbf{W}_0]$, but pruning the model (*i.e.*, increase the sparsity of $\mathbf{M}_e$) $t \times 100\%$ further cannot produce a winning ticket. In our experiments, we set $t$ to be 0.01.

### 3.2 Verification Framework for Extremely Sparse Winning Tickets

A ownership verification framework for extremely sparse winning tickets can be formulated as a tuple $\mathcal{V} = (ME, WE, F, V, I)$, each item of which is a process:

- A *mask embedding* process $ME(\mathbf{M}_0, \mathbf{s})$ (optionally) for sparse masks. $\mathbf{s}$ is an optional string that can be encoded into masks. The output of this process is a new mask $\mathbf{M}$. $\mathbf{M}$ can either be a mask with $\mathbf{s}$ embedded or contains other structural information that is useful for ownership verification. We call the verification method with $ME(\mathbf{M}_0, \mathbf{s})$ enabled a "mask-based" method.

- A *weight embedding* process $WE(\mathbf{D}_{\mathrm{tr}}, \mathbf{T}, \mathbf{s}, \mathbb{N}[\cdot], L, \mathbf{W}_0, \mathbf{M})$, which is a learning process for the lottery ticket model. $\mathbf{D}_{\mathrm{tr}} = \{\mathbf{x}, y\}$ is the training dataset, $\mathbf{T} = \{\mathbf{T}_x, \mathbf{T}_y\}$ is an optional trigger set provided to the training process, $\mathbf{s}$ is an optional signature for the weights embedding process. $L$ is the loss function for model training (usually the cross-entropy loss), $N[\cdot]$ defines the model structure, $\mathbf{W}_0$ is the initialization of weights, and $\mathbf{M}$ is the sparse mask for the model. The output of $E$ is a model $\mathbb{N}$ with sparse weights $\mathbf{W} \odot \mathbf{M}$ where $\mathbf{W}$ represents the trained weights. The

trigger set $\mathbf{T}$ (or/and) the signature $\mathbf{s}$ are embedded in $\mathbf{W} \odot \mathbf{M}$ after this process and can be verified with the verification process introduced next.

- A *fidelity evaluation* process $F(\mathbb{N}[\cdot], \mathbf{W}, \mathbf{M}, \mathbf{D}_{\text{te}}, \mathcal{A}_f, \epsilon_f)$ is to evaluate whether the performance discrepancy of model $\mathbb{N}[\cdot]$ is less a pre-defined threshold $\epsilon_f$, *i.e.,* $|\mathcal{A}(\mathbb{N}[\mathbf{W}, \mathbf{M}], \mathbf{D}_{\text{tr}}) - \mathcal{A}_f| < \epsilon_f$, in which $\mathcal{A}(\cdot, \cdot)$ is the inference performance on the test dataset $\mathbf{D}_{\text{te}}$, and $\mathcal{A}_f$ is a target inference performance associated with the model.

- A *verification* process $V(\mathbb{N}[\cdot], \mathbf{W}, \mathbf{M}, \mathbf{T}, \mathbf{s}, \epsilon_s)$ checks whether the sparse mask $\mathbf{M}$ or the trigger set $\mathbf{T}$ can be successfully verified for a given model $\mathbb{N}[\cdot]$. For the mask-based methods, the process is to check if $\mathbf{M}$ and $\mathbf{s}$ matches by evaluating $N[\cdot, \mathbf{M}]$ on $\mathbf{D}_{\text{te}}$ to see if the performance gap is smaller than a pre-defined threshold $\epsilon_s$, and(or) extract information in $\mathbf{M}$ and compare it with $\mathbf{s}$. For the trigger set-based methods, an inference is first executed on the trigger set images $\mathbf{T}_x$, and then the prediction will be compared with trigger set labels $\mathbf{T}_y$ to see if the false detection rate is lesser than a threshold $\epsilon_s$ [34].

- An *invert* process $I(N[\mathbf{W}, \mathbf{M}], \mathbf{T}, \mathbf{s})$ exists and will enable a successful ambiguity attack [34] if: a) a set of new trigger set $\mathbf{T}'$, a new signature $\mathbf{s}'$, or a new mask $\mathbf{M}'$ can be reverse-engineered for the given mask $\mathbf{M}$ and weights $\mathbf{W}$. b) the forged $\mathbf{T}', \mathbf{s}', \mathbf{M}'$ can be verified with respect to $\mathbf{M}$ and $\mathbf{W}$. c) the fidelity evaluation outcome $F(\mathbb{N}[\cdot], \mathbf{W}, \mathbf{M}', \mathbf{D}_{\text{te}}, \mathcal{A}_t, \epsilon_f)$ remains True.

The high-level definitions above are general and can work with any concrete implementation. We will introduce several methods that use sparse structural information in the next following sections.

### 3.3 Structural Information As Signatures

Our motivation is originated from the nature of winning tickets that the sparse structure of winning tickets is critical to their performance. As the sparse masks found by IMP outperform other pruning methods by clear margins [5], incorrect masks will lead to degraded test accuracy. In the next few sections, we demonstrate how to use the sparse structure of extreme tickets, *i.e.,* both the sparse masks and weights, to perform ownership verification under different verification schemes. The ownership verification can be performed in two different scenarios: (a) protecting the sparse masks of the extremely sparse winning tickets; and (b) protecting the trained extremely sparse winning tickets.

### 3.3.1 Protecting the Sparse Masks: Splitting Signature from Sparse Model

Such sparse masks play a crucial role in achieved outstanding generalization [1] and transferability [24, 25], and thus draws our attention to prevent them from being illegal distributed or used. Given a fixed initialization, correct masks are essential for training the extremely sparse winning tickets to match the performance of the dense network. If we split the sparse masks into two parts, neither part is intact and correct so neither can be trained to match the performance of the dense model with the given initialization. Recall the mechanism of one lock can be unlocked by one key generally, we adopt the concept of keys and locks and propose a new ownership verification method for the masks of extremely sparse winning tickets.

Denote the sparse mask of extremely sparse winning ticket by $\{\mathbf{M}_l\}_{l=1}^N$ and the weights by $\{\mathbf{W}_l\}_{l=1}^N$, where $N$ is the number of layers. To sparsify a model, $\{\mathbf{M}_l\}_{l=1}^N$ is applied to the model's weight $\{\mathbf{W}_l\}_{l=1}^N$ by conducting an element-wise product ($\{\mathbf{W}_l \odot \mathbf{M}_l\}_{l=1}^N$). Our goal is to find *key masks i.e.,* sub-masks $\{\mathbf{M}_l^s\}_{l=1}^N$ that contain as few elements as possible while the performance of the sparse network with the *locked masks*, *i.e.,* the remaining masks, degrade as much as possible. Meanwhile, fewer elements in key masks reduce the cost of storing, distributing, and using the key masks.

We next describe the algorithms needed to discover key masks. An algorithm is used to split the masks of extremely sparse winning tickets ($\{\mathbf{M}_l\}_{l=1}^N$) into key masks $\{\mathbf{M}_l^s\}_{l=1}^N$ and locked masks under the constraint of $\text{rspar}(\{\mathbf{M}_l^s\}_{l=1}^N, \{\mathbf{M}_l\}_{l=1}^N) < n_s$. $n_s$ is a hyper-parameter controlling the relative sparsity of the key masks. Score functions are used to decide which part should be split into the key masks. The pipeline is described in Algorithm 1.

We study several score functions in our experiments: 1) One-Shot Magnitude (OMP): the absolute values of each weight; 2) Edge-Weight-Product (EWP) [35] which measures the importance of paths from models' input to output. The EWP score is defined as the multiplication of weights along the paths; 3) Edge betweenness centrality (Betweenness). The edge betweenness centrality measures the importance of each edge inside a graph. For convolutional layers, we define the weight of each "edge" to be the summation of absolute values of each element; and 4) random scoring.

**Algorithm 1:** Splitting Key Masks

**input** : A sets of masks $\mathbf{M} = \{\mathbf{M}_l\}_{l=1}^N$, initialization weights $\mathbf{W} = \{\mathbf{W}_l\}_{l=1}^N$, number of non-zero elements $n$, and a function $\texttt{score}(\cdot)$ for scoring.

**output** : Key masks $\{\mathbf{M}_l^s\}_{l=1}^N$ and locked masks $\{\mathbf{M}_l - \mathbf{M}_l^s\}_{l=1}^N$

1 Derive the score matrices by applying $\texttt{score}(\cdot)$ over $\{\mathbf{W}_l \odot \mathbf{M}_l\}_{l=1}^N$ and get $\{\mathbf{S}_l\}_{l=1}^N$

2 Set the values of entries in $\{\mathbf{S}_l\}_{l=1}^N$ to negative infinity if the corresponding entries at the same position in $\{\mathbf{W}_l \odot \mathbf{M}_l\}_{l=1}^N$ is zero (*i.e.*, already pruned).

3 Calculate the $n^{\text{th}}$ largest number across the score matrix $\{\mathbf{S}_l\}_{l=1}^N$ and record it as $T$.

4 Set $\mathbf{M}_l^s \leftarrow I_{\mathbf{M}_l > T}$ and let the key masks be $\{\mathbf{M}_l^s\}_{l=1}^N$. The comparison between $\mathbf{M}_l$ and $T$ ($\mathbf{M}_l > T$) is performed element-wise.

### 3.3.2 Protecting the Trained Tickets: Embedding Signature into Sparsity Masks

Another scenario is to protect the trained extremely sparse winning ticket since a superior performance on certain large-scale datasets usually comes with a huge economic and ecological cost. Although directly splitting the masks provides a solution to the ownership verification problem, it has some drawbacks. It delivers extra cost to users since they need to recover the masks. Such a method is also intrusive and requires additional responsibility from the users' side for storing the key masks safely. To render the extreme tickets capable of self-verification and free of key masks , we propose a novel pruning method that is able to "absorb" secret information (*e.g.*, signatures) into models' sparse masks. The core concept is to enforce the sparsity masks to follow certain "0-1" patterns, which can be extracted from masks and further decoded back to the original form of information.

A function $\texttt{encode}(\cdot)$ is used to transform a string $\mathbf{s}$ into a matrix $\mathbf{M}_s \in \{0,1\}^{d_1 \times d_2}$ which we call *signature mask*. Our goal is to embed $\texttt{encode}(\mathbf{s})$ into the sparsity masks $\{\mathbf{M}_l\}_{l=1}^N$. One critical question is where to embed the signature mask $\mathbf{M}_s$. Empirically, low-level convolutional layers are less sparser, which means they are more unlikely to be pruned. Therefore, information embedded in the low-level convolutional layers is more difficult to be removed if using the pruning method. Based on such observation, we decide to embed $\mathbf{M}_s$ in low-level convolutional layers. To minimize mask changes, we first find a region in $\{\mathbf{M}_l\}$ with the highest similarity with $\mathbf{M}_s$ and tune the sub-mask of that region. For masks that have a dimension of two, we directly replace the region with $\mathbf{M}_s$; for masks that have a dimension of more than two, we raise their dimension by using random connections. Our detailed workflow is shown in Algorithm 2.

The choices of function $\texttt{encode}(\cdot)$ are various but there is one common choice in our daily life: QR code [36]. QR code has multiple advantages: 1) QR code is naturally seen as a pattern with only zeros and ones; 2) QR code has the ability to correct the error if the code is dirty or damaged. For example, the H correction level can tolerate up to 30% of error [36]; 3) QR codes can be small in size which can be easily fit into sparse masks. The size of the QR code generated can be as small as $21 \times 21$ while the numbers of channels in convolutional kernels in deep learning models are typically greater than 21, showing an abundant space for fitting the QR code in inside models' sparsity masks; and 4) The QR code **without** the finder, alignment and version patterns are imperceptible when fitted into sparse masks since there are no "regular" patterns left. Based on these merits, we choose $\texttt{encode}(\cdot)$ to be the QR code generation function. Specifically, the $\texttt{encode}$ function we use will return a QR code without finder, alignment, and version patterns. When extracting the code, the above patterns will be added back for decoding the credential information behind the QR code.

**Algorithm 2:** Embed Signature Into Sparse Masks

**input** : A set of masks $\mathbf{M} = \{\mathbf{M}_l\}_{l=1}^N$, a signature $\mathbf{s}$

**output** : A set of masks with signature embedded $\{\mathbf{M}_l^e\}_{l=1}^N$

1 Calculate $\mathbf{M}_s \leftarrow \texttt{encode}(\mathbf{s})$.

2 Squeeze each $\mathbf{M}_l$ into a two-dimensional matrix $\mathbf{M}_l^f$ by setting $(\mathbf{M}_l^f)_{ij} = \mathbb{I}_{\|(\mathbf{M}_l)_{ij}\|_0 > 0}$.

3 Calculate the similarity (percentage of matched 0-1 patterns) between each $\mathbf{M}_l^f$ and $\mathbf{M}_s$ and name the one with the largest similarity $\mathbf{M}_{max}^f$.

4 Change the dimension of $\mathbf{M}_s$ and fit it into $\mathbf{M}_{max}^f$ to the region where the similarity is the largest.

### 3.4 Ownership Verification with Sparse Structural Information

Next, we propose three different verification schemes based on sparse structural information, as summarized in Table A9. Under our unique context, we further introduce a new *Add-on Attacks* which aims to create ambiguity against lottery verification by "recovering" several pruned weights and manipulating the sparsity patterns. More details can be found in Appendix A1.

**Scheme $\mathcal{V}_1$: Distribute the extreme tickets with key masks.** Scheme $\mathcal{V}_1$ is designed to protect the sparsity masks. We separate the sparsity mask $\mathbf{M}$ into two parts: $\mathbf{M}_l$ and $\mathbf{M}_s$, where l/s subscripts denote "large"/"small", respectively. The small mask is sparser than the large one, which is used as the key mask while the large counterpart is the locked mask. We apply these two masks on weights and get two separate parts $(\mathbf{W} \odot \mathbf{M}_l, \mathbf{W} \odot \mathbf{M}_s)$. Before re-training, legitimate users should merge the two weights by adding them up to recover the original sparse weights $\mathbf{W} \odot \mathbf{M}$. The ownership can be automatically verified by the inference performance since an incorrect provided mask-weight pair will deteriorate accuracies after re-training.

**Scheme $\mathcal{V}_2$: Embed signatures in sparse masks.** We apply the signature mask embedding method to embed credentials into the extreme ticket. Then we train the model and dispatch it to legitimate users. No further action is required at the users' side. For the verification process, one can use extract the signature from the sparse model and validate the ownership of the extreme tickets. Compared with Scheme $\mathcal{V}_1$, Scheme $\mathcal{V}_2$ is more user-friendly since no extra action is performed at the users' side. The application scenarios of Scheme $\mathcal{V}_1$ and $\mathcal{V}_2$ are also different: the latter focuses on protecting the trained weights. It also shows great defense ability towards removal and ambiguity attacks. However, this scheme works under the white-box verification setting only, which means that access to models' weights has to be assumed. To overcome that assumption, we combine Scheme $\mathcal{V}_2$ with a trigger set-based method and propose Scheme $\mathcal{V}_3$ in the next section.

**Scheme $\mathcal{V}_3$: Combining trigger set-based methods.** Scheme $\mathcal{V}_3$ is more sophisticated than Scheme $\mathcal{V}_2$ as a set of trigger images and labels are used during the (re-)training process. With the help of this trigger set, Scheme $\mathcal{V}_3$ is now capable of black-box verification. By using remote calls of service APIs, the owner can first probe and claim the ownership in a black-box regime and further request a white-box verification if the black-box mode has raised a red flag. The white-box verification part for Scheme $\mathcal{V}_3$ remains the same as Scheme $\mathcal{V}_2$.

## 4 Experiments

In this section, we will list the details of our experiments and show the results to prove the effectiveness of our ownership verification methods, as well as the robustness to removal attacks (*e.g.*, model pruning, fine-tuning, and add-on attacks) and ambiguity attacks (*e.g.*, fake paths, fake code).

**General Settings.** We use three networks architectures (ResNet-20, ResNet-18 and ResNet-50) and two benchmarks (CIFAR-10 [37] and CIFAR-100 [37]) in our experiments. For all experiments, we follow the same (re-)training and testing protocol. The optimizer we use is an SGD optimizer with a momentum factor of 0.9 and a weight decay factor of 1e-4 for ResNet-20 and ResNet-18, and 5e-4 for ResNet-50. We train the model for 182 epochs with an initial learning rate of 0.1, and we decay the learning rate at $91^{st}$ and $136^{th}$ epoch by 0.1. We use a late rewinding technique that rewinds the weight to the $3^{rd}$ checkpoint. More results on ResNet-50 are deferred to Appendix A2. Our experiments are run with 16 pieces of NVIDIA RTX GeForce 2080 Ti.

**Types of Attacks and Trigger Set.** We study two types of attacks: 1) removal attacks. This type of attack aims at removing the embedded watermarks from the model's weights or data. Available methods for removal attacks include pruning, which removes a proportion of parameters of the model, and fine-tuning, which performs training on new data for a few steps. Both methods can modify the weights and potentially make the watermark undetectable. 2) ambiguity attacks. This type of attack aims at confusing the verification schemes, *i.e.,* no one can tell which is the real watermark/signatures. This type of attack needs techniques like reverse engineering and does not have a certain form. We explore several attack methods in our paper.

The trigger set we use is the same as the trigger set used in [7], which contains abstract images that are different than the training images.

**New type of attack: Add-on Attacks.** Although the extremely sparse winning tickets are naturally robust to fine-tuning and pruning attacks due to their unique properties, the verification schemes based on the sparse structure will potentially suffer from another kind of attacks, i.e., trying to "recover"

some pruned weights and change the sparse structure of the extremely sparse winning tickets. We name it *add-on attacks*. Such a new attack type targets mask-based verification schemes and aims at creating ambiguity against verification.

We propose a pipeline defending against such attacks. We can first prune weights whose magnitudes are smaller than $t$. $t$ is known to the owner of the model since the owner has the authentic sparse masks. This can detect any noise with magnitude smaller than $t$, that the attackers add to the mask. For noises of moderate level, their magnitudes become comparable to with the benign weights, hence the prediction quality will be significantly degraded as the noise increases.

## 4.1 Finding Extreme Tickets

To find extreme tickets, multiple rounds of the train-prune-retrain process are usually required. Once the test performance of the currently trained model cannot match the performance of the dense model, we revert the pruning process back for one time and reduce the pruning ratio. The choices of pruning ratio of weights are $[0.2, 0.1, 0.05, 0.1]$. The results are shown in Table 1. We also report the pruning specification, which includes the remaining weights of the extreme tickets, as well as the pruning times for each pruning rate we choose.

Table 1: Performance of dense models and extremely sparse winning tickets, and the pruning specification. The performance are expressed in terms of the test accuracy of the dense model and the extremely sparse winning ticket. The pruning specification includes the proportion of remaining weight as well as the number of pruning with four different pruning ratios (in brackets).

| Model | Dataset | Performance | Pruning Specification |
|---|---|---|---|
| ResNet-20 | CIFAR-10 | 91.67%,91.66% | 19.369% (5,1,8,1) |
| ResNet-20 | CIFAR-100 | 66.36%,66.39% | 19.901% (6,0,4,7) |
| ResNet-18 | CIFAR-10 | 93.67%,93.60% | 1.236% (18,3,0,6) |
| ResNet-18 | CIFAR-100 | 72.44%,72.59% | 2.251% (17,0,0,0) |

Notice that ResNet-20 needs at least 15 times of pruning before we identify the extreme tickets, and for ResNet-18, such number increases to 17. Numerous pruning rounds exemplify the effort to find the extreme tickets and emphasizes the importance of protecting them.

## 4.2 Effectiveness of Different Schemes

**Scheme $\mathcal{V}_1$** To verify that extreme tickets without correct key masks will have degraded performance after retraining, we conduct experiments that directly re-train the models without the key masks on ResNet-20 and ResNet-18. Fig. 2 show the performance of extreme tickets after retraining. It can be seen that more "1"s in key masks can increase the performance divergence. Different splitting functions do not make an essential difference, showing that the choice of `score` functions is flexible.

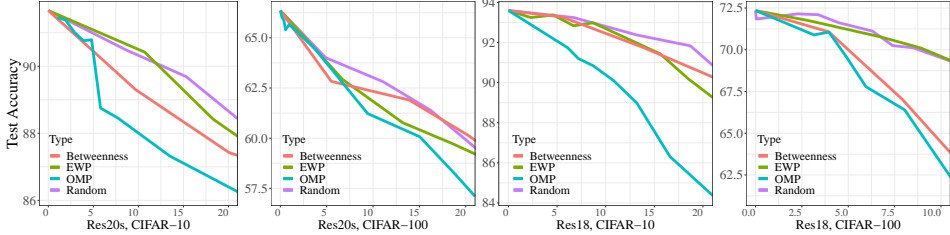

Figure 2: Effectiveness of Scheme $\mathcal{V}_1$: Re-training without key masks generated by four methods: Betweenness, EWP, OMP, Random (Paths). The $x$-axis is the relative sparsity w.r.t the extreme ticket.

**Scheme $\mathcal{V}_2$** To show that our proposal can embed information into models' sparse structures without significantly harming their performance, we conduct experiments to compare the performance before and after a signature string is embedded. Table 2 shows the test accuracy of two different models. We can see from the table that the performance of extreme tickets with the string embedded is only slightly lower than the original one, which proves that using Scheme $\mathcal{V}_2$ endows owners the ability to embed information at a little cost of performance.

**Scheme $\mathcal{V}_3$** We use a set of trigger images during the re-training of extremely sparse winning tickets under Scheme $\mathcal{V}_3$. The inference performance on the original task (*i.e.*, CIFAR-10 or CIFAR-100) should not be greatly affected with trigger sets enabled. Table 3 shows the inference performance on both the original images and the trigger set. We can see that the performance only drops 0.2% on CIFAR-10 and 0.97% on CIFAR-100 for ResNet-20, while the detection rates of the trigger set images are high (91.0% on CIFAR-10 and 90.0% on CIFAR-100). At the same time, the extreme tickets trained from CIFAR-10 and CIFAR-100 without a trigger set can only have trigger set accuracies of

Table 2: Effectiveness of Scheme $\mathcal{V}_2$: performance of extreme tickets after embedding QR codes. We study two different models and compare their inference performance. The performance after embedding and the performance drop are reported (in brackets).

| Model | Accuracy after embedding | |
|---|---|---|
| | CIFAR-10 | CIFAR-100 |
| ResNet-20 | 91.37% (↓ 0.29%) | 66.14% (↓ 0.25%) |
| ResNet-18 | 93.56% (↓ 0.11%) | 72.35% (↓ 0.24%) |

Table 3: Effectiveness of Scheme $\mathcal{V}_3$: ResNet-20 on CIFAR-10 and CIFAR-100. ESWT is the abbreviation of **E**xtremely **S**parse **W**inning **T**ickets. We re-train the two extremely sparse winning tickets with QR code embedded found with the trigger set enabled.

| Model | Test Accuracy | |
|---|---|---|
| | CIFAR-10 | CIFAR-100 |
| ESWT | 91.66% (16.0%) | 66.36% (0.0%) |
| ESWT + $\mathbf{M}_s$ + $\mathbf{T}$ | 91.46% (91.0%) | 65.39% (90.0%) |

16.0% and 0.0%, respectively. This suggests the Scheme $\mathcal{V}_3$ can work as expected, *i.e.,* perform well on the trigger set while not significantly harming the performance on the original dataset.

### 4.3 Robustness Against Removal Attacks

**Fine-tuning Attacks**  Fine-tuning the model can only change the values of weights while not changing the sparse structure of extreme tickets. As a consequence, Scheme $\mathcal{V}_2$ and $\mathcal{V}_3$ is resistant to fine-tuning attacks under the white-box verification setting.

For Scheme $\mathcal{V}_1$, users are required to provide the key masks to recover the correct masks and then re-train the extreme tickets. One key property we need to verify is that attackers cannot bypass the requirement of key masks by fine-tuning the model on a new dataset. To this end, we conduct transfer experiments described as follows: on CIFAR-100, we train the model with the locked mask generated on the extreme tickets identified on CIFAR-10; on CIFAR-10, we conduct a similar experiment with the locked mask from CIFAR-100. The results are shown in Table 4. From the table, we can see that even transferring the sparse mask cannot bypass the requirement of key masks. The performance gaps between the transferred model and the extreme tickets found on each set are greater than 3% on both datasets, much higher than the 1% criterion we set for matching performance. Such big gaps prove that the model after fine-tuning attacks is not useful in practice.

Table 4: Fine-tuning Attacks on Scheme $\mathcal{V}_1$: Transferring extreme tickets of ResNet-20 found on CIFAR-10/100. 10→100 means transferring from CIFAR-10 to CIFAR-100 and vice versa. The percentage inside brackets denotes the relative sparsity of the key mask w.r.t the extremely sparse winning ticket.

| Model | Test Accuracy | |
|---|---|---|
| | 10→100 | 100→10 |
| OMP (5%) | 59.80% | 87.66% |
| EWP (5%) | 60.27% | 88.21% |
| Betweenness (5%) | 59.61% | 87.22% |

Table 5: Pruning Attacks on Scheme $\mathcal{V}_3$: Performance of ResNet-20 on CIFAR-10/100 after pruning with different pruning ratios. The accuracy on CIFAR-10/100 are shown outside the brackets and the accuracy on trigger images are inside the brackets.

| Model | Accuracy | |
|---|---|---|
| | CIFAR-10 | CIFAR-100 |
| Original model | 91.46% (91.0%) | 65.39% (90.0%) |
| Pruning 5% | 91.33% (89.0%) | 64.78% (91.0%) |
| Pruning 10% | 90.66% (90.0%) | 62.96% (73.0%) |
| Pruning 20% | 87.86% (81.0%) | 50.14% (16.0%) |
| Pruning 50% | 33.04% (18.0%) | 8.56% (0.00%) |

We also have conducted experiments to study if Scheme $\mathcal{V}_3$ can resist fine-tuning attacks under black-box verification. We first retrain the extreme tickets under Scheme $\mathcal{V}_3$ on CIFAR-10/-100, and continue to fine-tune it on CIFAR-100/-10. The extreme tickets trained on CIFAR-10 can only achieve 61.59% test accuracy on CIFAR-100, and the extreme tickets on CIFAR-100 can only achieve 88.21% test accuracy on CIFAR-10. The strong bond between sparse structure (masks of extreme tickets) and datasets on which the extreme tickets we found brings performance drop when fine-tuning them on a new dataset, which devalues such attack and also highlight the robustness of the Scheme $\mathcal{V}_3$ against fine-tuning attacks.

**Model pruning**  Pruning the model under Scheme $\mathcal{V}_1$ is meaningless since pruning cannot recover the full masks. So we focus on Scheme $\mathcal{V}_2$ and $\mathcal{V}_3$ for model pruning attacks. Pruning the trained model leaves more "0" in the trained model, which might change the extracted QR code and makes it unable to decode. To study if our model can resist the pruning attack, we conduct experiments with different pruning methods (one-shot magnitude and random pruning) and different pruning ratios (5%, 10%, 20%, 30%, 50%).

We first examine our proposal for black-box verification (Scheme $\mathcal{V}_3$). In Table 5 we show the results of Scheme $\mathcal{V}_3$ against pruning attacks. The accuracy on trigger set images drops after the accuracy on the original dataset (CIFAR-10/CIFAR-100) has decreased considerably, which means that the

user-specific information cannot be removed without sacrificing its performance and demonstrates its resilience against pruning attack.

We then test our proposal for white-box verification (Scheme $\mathcal{V}_2$). In Table 6 we show the inference performance on original datasets after pruning, and also show the QR codes extracted from masks in Figure 3. We can see that the performance of the pruned model will degrade dramatically after 20% percent of one-shot magnitude pruning and 5% percent of random pruning. On the contrary, the QR code extracted from the ResNet-20 can be decoded even after 20% percent of one-shot magnitude pruning. Figure 4 shows the QR code extracted from ResNet-18. At the 5% pruning ratio, the string can be easily decoded into a readable string. At the 10% pruning ratio, the string can still be partly decoded, although the readability has been reduced. For the pruning ratio greater than 10%, the inference performance has significantly dropped, making it meaningless to conduct such attack.

Table 6: Inference performance of extremely sparse winning tickets on ResNet-20 and ResNet-18 after model pruning attacks under different pruning methods and pruning ratios. OMP stands for one-shot magnitude pruning. The numbers in brackets stand for the pruning ratios.

| Method (Percent) | Performance | |
|---|---|---|
| | CIFAR-10 | CIFAR-100 |
| Scheme $\mathcal{V}_2$ | 91.37% | 72.35% |
| OMP (5%) | 91.25% | 72.27% |
| OMP (10%) | 90.72% | 71.42% |
| OMP (20%) | 88.03% | 69.51% |
| OMP (30%) | 80.08% | 60.31% |
| OMP (50%) | 36.62% | 9.24% |
| Random Pruning (5%) | 60.87% | 58.23% |
| Random Pruning (10%) | 30.49% | 22.67% |
| Random Pruning (20%) | 11.95% | 3.23% |
| Random Pruning (30%) | 12.05% | 1.0% |
| Random Pruning (50%) | 10.00% | 1.0% |

Table 7: Summary of different types of ambiguity attacks. We show the specification of each attack, *i.e.*, the accessibility of each component to attackers, the attack methods, and the targeted schemes.

| Attack name | Attackers can access | How to attack | Attack Scheme |
|---|---|---|---|
| $fake_1$ | $\mathbf{W} \odot \mathbf{M}_l$ | Forge $\mathbf{W} \odot \mathbf{M}_s$ | Scheme $\mathcal{V}_1$ |
| $fake_2$ | $\mathbf{W} \odot \mathbf{M}$ | Add noise $\mathbf{W}_{noise} \odot \mathbf{M}_{noise}$ | Scheme $\mathcal{V}_2$ and $\mathcal{V}_3$ |
| $fake_3$ | $\mathbf{W} \odot \mathbf{M}$ and $\texttt{encode}(\cdot)$ | Replace $\mathbf{M}_s$ | Scheme $\mathcal{V}_2$ and $\mathcal{V}_3$ |

Table 8: Test accuracy and remaining weights after add-on attacks under different rates on ResNet-20, with the matching condition and the decode-ability of the QR code extracted from the masks.

| Add-on Rate | Test Accuracy (% $r_{remain}$) | Decode-able? | Match? |
|---|---|---|---|
| 0% | 91.53% (19.369%) | ✓ | ✓ |
| 0.5% | 91.04% (19.789%) | ✓ | ✓ |
| 1% | 90.23% (20.179%) | ✓ | ✗ |
| 2% | 86.64% (21.009%) | ✗ | ✗ |
| 5% | 79.49% (23.386%) | ✗ | ✗ |
| 10% | 71.06% (27.402%) | ✗ | ✗ |

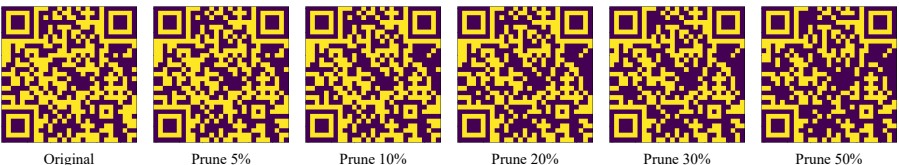

Original    Prune 5%    Prune 10%    Prune 20%    Prune 30%    Prune 50%

Figure 3: QR code extracted from ResNet-20 under pruning attacks with different ratios. The codes extracted under 5% and 10% pruning ratio can be easily decoded into readable strings "signature", and the code under 20% pruning ratio can be decoded into "sigiature" with tools at https://github.com/merricx/qrazybox/.

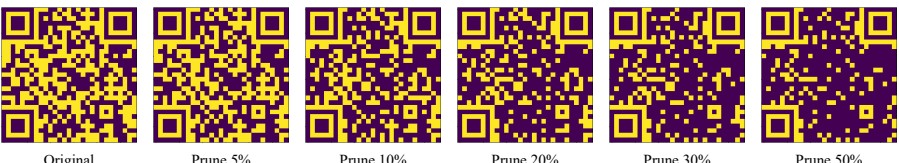

Original    Prune 5%    Prune 10%    Prune 20%    Prune 30%    Prune 50%

Figure 4: QR code extracted from ResNet-18 under pruning attacks with different ratios. The code extracted under 5% can be easily decoded into a readable string "signature", and the code under 10% pruning ratio can be decoded into "simçi@5re" with tools at https://github.com/merricx/qrazybox/.

## 4.4 Resilience Against Ambiguity Attacks

In this section, we will evaluate the robustness against ambiguity attacks summarized in Table 7.

**Scheme $\mathcal{V}_1$** (*fake$_1$*: Attackers can access $\mathbf{W} \odot \mathbf{M}_l$ only) The goal of *fake$_1$* is to forge a new key mask $\mathbf{M}'_s$ with new underlying weights $\mathbf{W}'$. As the attacker has no prior information on $(\mathbf{W} \odot \mathbf{M}_s)$, the forging process can only be performed randomly. From Figure 5 we can see that such an attack method is not practical as the performance gap is much greater than $\epsilon_f (= 1\%)$ after using random key masks. For example, if we adopt OMP as the scoring function to construct the key masks, we only need a key mask of around 10% relatively sparsity to make the model resistant to random attacks.

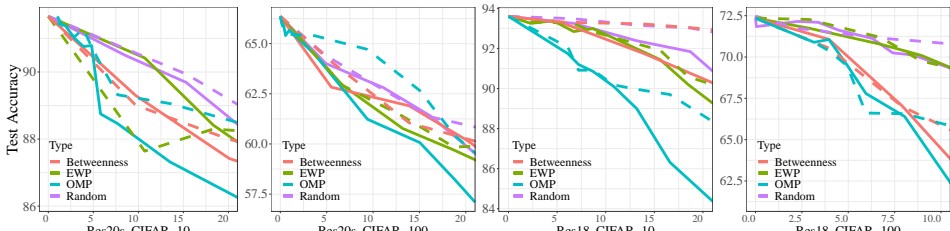

Figure 5: Random attacks on Scheme $\mathcal{V}_1$. The $x$-axis is the relative sparsity of the key masks. The solid/dashed lines represent the performance **before/after** random attacks.

**Scheme $\mathcal{V}_2$ and $\mathcal{V}_3$** (*fake$_2$*: Attackers can access $\mathbf{W} \odot \mathbf{M}$ but not knowing $\texttt{encode}(\cdot)$) One might use add-on attacks and try to "contaminate" the information we embed in the sparse mask. Specifically, we randomly add noises to the position where the weights are pruned. We test with add-on rates ranging from 0% to 10% since a 10% efficiency gap will diminish the value of attacking the model. The results are shown in Table 8 and Figure 6. From the table, we can see that introducing 1% of noise to the trained model will un-match the attacked model (*i.e.,* the performance gap becomes greater than 1%). For the add-on rates smaller than 1%, the QR code embedded in the sparse mask can be normally decoded into a normal string. Such results prove that both Scheme $\mathcal{V}_2$ and $\mathcal{V}_3$ are resistant to attack *fake$_2$*.

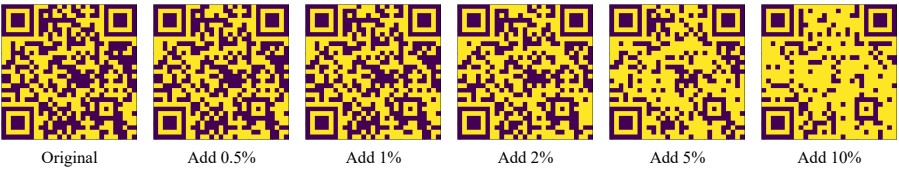

| Original | Add 0.5% | Add 1% | Add 2% | Add 5% | Add 10% |

Figure 6: Visualization of QR code extracted and processed under different add-on rates.

(*fake$_3$*: Attackers can access $\mathbf{W} \odot \mathbf{M}$ and $\texttt{encode}(\cdot)$) If the attacker knows about the $\texttt{encode}$ function for generating the $\mathbf{M}_s$, a similar but fake signature mask $\mathbf{M}_s'$ which contains a different signature can be generated in the same way. However, as shown in Figure 1, without the finder, alignment, and version patterns, one can hardly tell which part belongs to a QR code. Even if the attacker knows the position where the code is embedded (namely an insider attack [34]), directly replacing the embedded region with a new signature mask $\mathbf{M}_s'$ and $\mathbf{W}'$ (noise) will also considerably degrade the performance of the attacked model since a large amount of "incorrect" weights are introduced. For example, for ResNet-20 on CIFAR-10, the test accuracy of the attacked model will drop from 91.37% to 57.00%, which is nearly a 50% degradation in performance. Such a big loss shows that it is infeasible to perform the insider attack.

## 5  Conclusion and Discussion of Broad Impact

LTH offers superior sparse models through burdensome explorations, serving as an intriguing yet expansive solution for resource-constrained applications. It motivates the necessity of protecting the copyright of these precious winning tickets. We investigate a brand new verification technique by leveraging the sparse structural information, which embeds signatures into lottery tickets' typologies. Extensive results verify our proposal's effectiveness and robustness against diverse malicious attacks.

This work is scientific in nature and should bring positive societal impacts. Note that every second, giant and start-up companies have invested billions of dollars to identify superior yet light-weight compact deep neural networks virtually. We believe our new *lottery verification* mechanism can assist both industry and academia in defending their interests from illegal distribution or usage.

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
