## A1   More Methodology Details

**More of ownership verification schemes.**   Table A9 summarizes our proposed ownership verification regimes. There are five different phases in each of our schemes: 1) *Ticket finding*: finding the

extremely sparse winning tickets. Multiple rounds of the train-prune-retrain process are involved in this phase for finding the extremely sparse winning tickets; 2) *Pre-Process*: pre-process the extremely sparse winning ticket for applying each scheme. For example, we need to construct the key masks if using the Scheme $\mathcal{V}_1$; 3) *Re-training*: this process is unique for the winning tickets that we will train the extremely sparse winning ticket again to match the performance of the dense model; 4) *Inference*: the inference process is to perform an inference process on the test dataset; 5) *Validation*: This process is to validate the ownership of the (trained/untrained) extremely sparse winning ticket.

Table A9: Summary of different ownership verification schemes. The re-training phase can be either done by the ticket owner or the legitimate users.

| | Scheme $\mathcal{V}_1$ | Scheme $\mathcal{V}_2$ | Scheme $\mathcal{V}_3$ |
|---|---|---|---|
| Ticket Finding | No additional technique | No additional technique | No additional technique |
| Pre-Process | Split key masks and locked masks Distribute both the masks | Calculate $\mathbf{M}_s$ using $\texttt{encode}(\cdot)$ Embed $\mathbf{M}_s$ into $\mathbf{M}$ and distribute | Calculate $\mathbf{M}_s$ using $\texttt{encode}(\cdot)$ Embed $\mathbf{M}_s$ into $\mathbf{M}$ and distribute |
| Re-training | Recover the masks | No additional technique | Training with the trigger set $\mathbf{T}$ |
| Inference | Keys masks are required Slight overhead for recovering the masks | No additional technique | No additional technique |
| Validation | Auto-verified by performance | Extract $\mathbf{M}_s$ and decode | Extract $\mathbf{M}_s$ and decode Inference on trigger set $\mathbf{T}$ |

## A2 More Experimental Results

**Extremely sparse winning tickets on ResNet-50.** On CIFAR-10, the remaining weights of the extremely sparse winning ticket is 13.19% (pruning specification: (7,1,6,0)) while the performance is 94.38% (0.04% drop). On CIFAR-100, the proportion of remaining weights of the extremely sparse winning ticket is 43.926% (pruning specification: (2,3,0,6)) while the performance is 75.84% (0.03% drop). On ImageNet, the proportion of remaining weights of the extremely sparse winning ticket is 16.97%, and the performance is 75.97% (0.01% higher).

**Extremely sparse winning tickets on VGG-16.** On CIFAR-10, the proportion of the remaining weights of the extremely sparse winning ticket is 1.44%, while the performance is 93.10% (0.04% higher). On Tiny-ImageNet, the proportion of the remaining weights of the extremely sparse winning ticket is 6.81%, while the performance is 58.12% (0.19% higher).

**Scheme $\mathcal{V}_1$ on ResNet-50.** Figure A7 shows the results of retraining the extremely sparse winning tickets without key masks. Multiple scoring functions (OMP, EWP, Random) are explored. It can be seen from the graph that on CIFAR-10, we need key masks with an approximately 15% relative sparsity to create a 1% performance gap, while on CIFAR-100, we need key masks with a relative sparsity of 5% approximately. ResNet-50 has greater model capacity than ResNet-20 and ResNet-18, so it is reasonable that we need more elements removed to reduce the performance significantly.

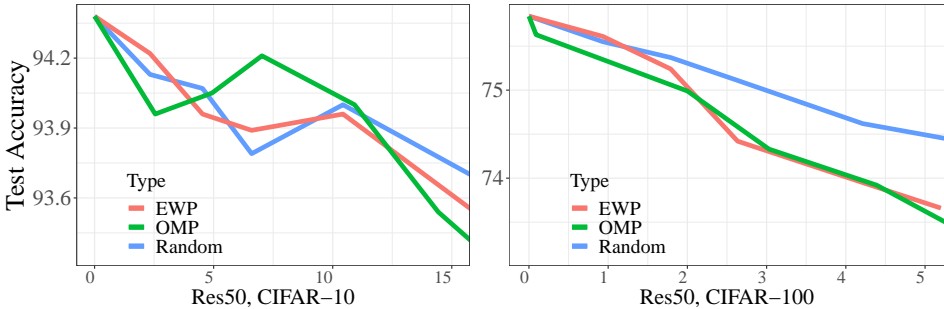

Figure A7: Effectiveness of Scheme $\mathcal{V}_1$: Re-training without key masks generated by three methods: EWP, OMP, Random. The $x$-axis is the relative sparsity w.r.t the extreme ticket.

On ImageNet, the performance of the retrained model is 75.39% when the relative sparsity is 0.4%, and the performance is 72.88%, which is nearly 3 percent lower when the relative sparsity is 5%. It proves that our Scheme $\mathcal{V}_1$ can work on large-scale datasets.

**Random ambiguity attacks on ResNet-50 under scheme $\mathcal{V}_1$.** Figure A8 shows the results of using random key masks for retraining the extremely sparse winning ticket for ResNet-50 on CIFAR-10 and CIFAR-100. It can be clearly seen from the graph that the random key masks will not contribute to recovering the performance of the trained model and even harm the test accuracy under some circumstances. On ImageNet, the accuracy of recovering masks with random connections is 75.32% and 74.57% when the relative sparsity is 0.4% and 5%, respectively. The performance gaps, which can be seen easily from the graphs and numbers, have demonstrated the robustness of Scheme $\mathcal{V}_1$ against the ambiguity attack.

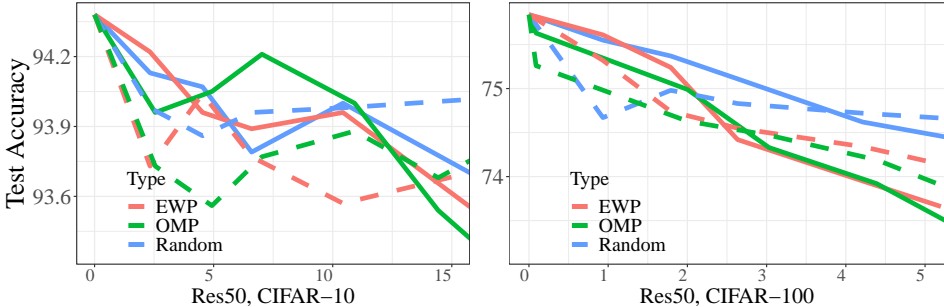

Figure A8: Random attacks on Scheme $\mathcal{V}_1$ on ResNet-50. The $x$-axis is the relative sparsity of the key masks. The solid/dashed lines represent the performance **before/after** random attacks.

**Scheme $\mathcal{V}_1$ on VGG-16.** On CIFAR-10, the performance of the retrained model without key masks is 88.63% when the relative sparsity of the key masks is 8%, and the performance after recovering with random connections is only 91.96%. On Tiny-ImageNet, the performance of the retrained model without key masks/with random key masks is 48.97%/52.86%. These results show the effectiveness and robustness of our Scheme $\mathcal{V}_1$.

**Scheme $\mathcal{V}_2$ and $\mathcal{V}_3$ on VGG-16.** We further examine the effectiveness and the robustness of the Scheme $\mathcal{V}_2$ and $\mathcal{V}_3$. The QR code embedded we put in the sparse mask of VGG-16 can still be partly decoded when the pruning ratio is 10%, while the test accuracy is 57.26% after pruning (0.7% lower) on Tiny-ImageNet. As for the Scheme $\mathcal{V}_3$, the test accuracy on Tiny-ImageNet decreases to 56.44% (over 1.5% lower) after pruning 20% of the trained model while the test accuracy on the trigger set is still 100%. All these phenomena show the effectiveness and robustness of our Scheme $\mathcal{V}_2$ and $\mathcal{V}_3$ on VGG-16.