# OpenReview forum: "You are caught stealing my winning lottery ticket! Making a lottery ticket claim its ownership"
_NeurIPS.cc/2021/Conference — NeurIPS 2021 Poster_

### Official Review · Reviewer_s6wM · 2021-07-10

**Rating:** 7
**Confidence:** 4

**Summary:**

Finding a special sparse subnetwork (winning ticket) from an overparameterized network is an expensive process. Due to the cost involved in obtaining this ticket, its owner might be interested in protecting it. This paper proposes graph-based signatures to perform lottery verification. Their method can work in both white-box and black-box settings. They show that due to the extremely sparse structure of the winning ticket, it can partly claim its ownership, without any specific protection mechanism. They give a mask embedding method to embed ownership signatures in the subnetwork. They claim such signature is robust to pruning, fine-tuning attacks. These claims are backed by experiments on ResNet models.

**Limitations And Societal Impact:**

Experiments are on one type of architecture (ResNets). I wonder if the claims would generalize to other architectures/ datasets/ tasks as well. Another thing that concerns me, is what are the odds that two people end up with similar pruned network, when they train+prune independently but use same open source code(random seed) and dataset?  How will such a situation be handled?


**Main Review:**

This paper gives novel methods for ownership verification using the structural information of the pruned network. In contrast, most of the existing works are based on watermarking either the weights of the model or watermarking labels of a trigger set. One of the main ideas is based on the fact that  pruning/changing an extremely sparse network will lead to performance degradation, so the structure/mask of an extreme sparse network can be used for ownership verification. Overall, the methods are technically sound and backed by decent experiments.

In context of current deep networks, pruning is very important to reduce training/inference costs. Pruning is expensive task but it is done once and the resultant subnetwork can be retrained again and again or used for inference. Hence it makes the protection of the subnetwork very important. Methods proposed in this paper are useful in such applications and their reliance on structural information seems to make them a better choice.

The paper is well written. I think some detailed background on ownership verification, protecting the model etc. can be very useful in appreciating the results.

**Time Spent Reviewing:**

3

---

> ### Author Response · Authors · 2021-08-10
> **Response to Reviewer s6wM**
>
> We thank the reviewer for appreciating our ideas and experiments results.
>
>
> **[Cons 1: Detailed Background needed]**
>
> We will include more detailed descriptions of the background knowledge in our final version, such as the definition of trigger sets and different attacks.
>
> **[Cons 2: More experiments on extra datasets and architectures.]**
>
> We have conducted multiple experiments with ResNet-50 on ImageNet (larger datasets), and with VGG-16 (various architectures) on CIFAR-10 and Tiny ImageNet (larger datasets). The performance of extreme winning tickets and their sparsity are presented in the below tables.  From the results we can see that our schemes can work very well with extra architectures and datasets, showing a great generality. Specifically, the performances of trained models without key masks have dropped considerably (over 3%) with a relative sparsity of less than 8%. Scheme 2 and Scheme 3 can also show the great ability of protection on VGG-16, indicating the effectiveness of our framework.
>
>
> | Architecture + Dataset | Baseline | Test Accuracy | Sparsity |
> | :-: | :-: | :-: | :-:|
> |ResNet-50 + ImageNet | 75.96% | 75.97% | 83.03% |
> | VGG-16 + CIFAR-10 | 93.06% | 93.10% | 98.56% |
> | VGG-16 + Tiny ImageNet | 57.93% | 58.12% | 93.19% |
>
>
> ResNet-50 + ImageNet with Scheme 1 (OMP):
>
> | Relative Sparsity | Accuracy | Add-back Accuracy |
> | :-: | :-: | :-: |
> | 0% | 75.97% | 75.97%|
> | 0.4% | 75.39% | 75.32%|
> | 5% | 72.88% | 74.57%|
>
>
> VGG + CIFAR-10 + Scheme 1 (OMP):
>
> | Relative Sparsity | Accuracy| Add-back Accuracy |
> | :-: | :-: | :-: |
> | 0% | 93.10% | 93.10% |
> | 8% | 88.63% | 91.96% |
>
>
> VGG + Tiny ImageNet + Scheme 1 (OMP):
>
> | Relative Sparsity | Accuracy| Add-back Accuracy |
> | :-: | :-: | :-: |
> | 0% | 58.12% | 58.12% |
> | 8% | 48.97% | 52.86% |
>
> VGG + Tiny ImageNet + Scheme 2:
>
> | Pruning Ratio | Accuracy| Decode-able? |
> | :-: | :-: | :-: |
> | 0% | 57.95% | Yes |
> | 5% | 57.82% | Yes |
> | 10% | 57.26% | Partly |
>
>
> VGG + Tiny ImageNet + Scheme 3:
>
> | Pruning Ratio | Accuracy | Trigger-set Accuracy |
> | :-: | :-: | :-: |
> | 0% | 57.93% | 100% |
> | 5% | 57.83% | 100% |
> | 10% | 57.56% | 100% |
> | 20% | 56.44% | 100% |
> | 50% | 8.24% | 31% |
>
> **[Cons 3: Problems of two similar pruned networks]**
>
> Thanks. Actually, It has been recently observed [r1] that even if using the same initialization and pruning method, as long as the SGD orders are different, the resultant sparse networks will have **largely dissimilar sets** of pruned weights. It is very unlikely that the SGD orders are all the same due to different facilities, software, and environments being used, therefore the chance to "accidentally shoot" benign sparse masks is empirically very low.
>
> Even if such a collision did happen, both Scheme 2 and Scheme 3 of our framework are still effective since user-designed and unique information can be embedded in the mask.
>
> [r1] ICLRW’21 - “Studying the Consistency and Composability of Lottery Ticket Pruning Masks”

---

### Official Review · Reviewer_w6uB · 2021-07-12

**Rating:** 8
**Confidence:** 4

**Summary:**

This paper studies the intersection of two frontiers: machine learning IP protection, and lottery ticket model. Since lottery ticket is resource-consuming to find and practically useful, the authors proposed lottery verification to identify a found ticket’s ownership.

**Limitations And Societal Impact:**

The impact is positive in general. The authors discussed in last section.

**Main Review:**

This is a very novel paper with solid execution. Details comments follow:

Pros
-	The problem studied in this paper is brand-new and the authors’ idea is refreshing. High quality sparse masks are very useful but more expensive to obtain than training the dense model, so protecting its IP is an important yet underexplored topic.
-	Besides common approach like “trigger set”, the protection of sparse masks can benefit further from its sparse structural information, offering strong promise of mask IP protection. The authors further studied how to embed signatures in masks, that can be extracted and decoded even after pruning or fine-tuning attacks
-	The authors conducted a particularly thorough empirical study on several verification schemes, i.e., separate masks, embedding signatures, and embedding signatures with trigger set.
-	They also consider attack types including removal, add-on and fine-tuning, covering possible manipulations of the sparse mask.
-	The protection results look strong on relatively large networks such as Res50. It is encouraging to see one relatively simple defense could resist all attacks considered.
-	Paper is well-written and easy to follow.
-	All codes were submitted in the supplement, seemingly easy to read and easy to use. The authors’ efforts are appreciated.

Cons
-	Experiments only on CIFR-10/100. While that’s fine as proof-of-concept, since their main argument is rooted in the expensiveness of finding tickets, showing that their method can scale up to larger datasets would be very helpful.
-	All attacks in this paper assumed stealing the mask associated with the original model, but that seems not necessary. Can knowledge distillation “steal” the function from the sparse tickets using another compact model?
-	Only limited transfer experiments are reported. How about the pretrain-then-transfer cases, e.g. in [25], can that mask be protected?


**Time Spent Reviewing:**

5

---

> ### Author Response · Authors · 2021-08-10
> **Response to Reviewer w6uB**
>
> We thank this reviewer for appreciating our novelty and experiment results, as well as acknowledging our contribution to the community.
>
> **[Cons 1: Experiments on larger datasets, i.e., Tiny-ImageNet and ImageNet]**
>
> We have conducted multiple experiments with ResNet-50 on ImageNet (larger datasets) and with VGG-16 (various architectures) on CIFAR-10 and Tiny ImageNet (larger datasets). The performance of extreme winning tickets and their sparsity are presented in the below tables.  From the results, we can see that our schemes can work very well with extra architectures and datasets, showing great generality. Specifically, the performances of trained models without key masks have dropped considerably (over 3%) with a relative sparsity of less than 8%. Scheme 2 and Scheme 3 can also show the great ability of protection on VGG-16, indicating the effectiveness of our framework.
>
>
> | Architecture + Dataset | Baseline | Test Accuracy | Sparsity |
> | :-: | :-: | :-: | :-:|
> |ResNet-50 + ImageNet | 75.96% | 75.97% | 83.03% |
> | VGG-16 + CIFAR-10 | 93.06% | 93.10% | 98.56% |
> | VGG-16 + Tiny ImageNet | 57.93% | 58.12% | 93.19% |
>
>
> ResNet-50 + ImageNet with Scheme 1 (OMP):
>
> | Relative Sparsity | Accuracy | Add-back Accuracy |
> | :-: | :-: | :-: |
> | 0% | 75.97% | 75.97%|
> | 0.4% | 75.39% | 75.32%|
> | 5% | 72.88% | 74.57%|
>
>
> VGG + CIFAR-10 + Scheme 1 (OMP):
>
> | Relative Sparsity | Accuracy| Add-back Accuracy |
> | :-: | :-: | :-: |
> | 0% | 93.10% | 93.10% |
> | 8% | 88.63% | 91.96% |
>
>
> VGG + Tiny ImageNet + Scheme 1 (OMP):
>
> | Relative Sparsity | Accuracy| Add-back Accuracy |
> | :-: | :-: | :-: |
> | 0% | 58.12% | 58.12% |
> | 8% | 48.97% | 52.86% |
>
> VGG + Tiny ImageNet + Scheme 2:
>
> | Pruning Ratio | Accuracy| Decode-able? |
> | :-: | :-: | :-: |
> | 0% | 57.95% | Yes |
> | 5% | 57.82% | Yes |
> | 10% | 57.26% | Partly |
>
>
> VGG + Tiny ImageNet + Scheme 3:
>
> | Pruning Ratio | Accuracy | Trigger-set Accuracy |
> | :-: | :-: | :-: |
> | 0% | 57.93% | 100% |
> | 5% | 57.83% | 100% |
> | 10% | 57.56% | 100% |
> | 20% | 56.44% | 100% |
> | 50% | 8.24% | 31% |
>
>
>
> **[Cons 2: Can knowledge distillation “steal” the function from the sparse tickets using another compact model?]**
> Great question. It is possible. However, we believe defending against model stealing using knowledge distillation is another parallel direction of protecting model IP that has been addressed recently, too [r1].  Our work is flexible and can seamlessly work together with their defense methods.
>
> Besides, it has been observed that when the teacher and student models have too large capacity differences (in our case, the teacher is the dense model and the student is a very compact one), the knowledge distillation is less effective [r2]. The design of compact students may also take extra effort.
>
> **[Cons 3: Pretrain-then-transfer experiments?]**
> Note that we follow the same pipeline in [25] that (i) identify extremely sparse winning tickets on the source task; (ii) the transfer found tickets to the target task. Technically, there should be no difference if we change the source task to the pre-training task since our verification framework is highly generalizable. To further address your concern, we first find extremely sparse winning tickets on the supervised pre-training task (i.e., ImageNet with ResNet-50). Then we transfer the identified ticket to CIFAR-10, with our Scheme 1 enabled. The baseline performance is 95.94% for reference. When the key mask has a relative sparsity of 5%, the performance drops 0.21%, and when the key mask has a relative sparsity of 10%, the performance drops 0.38%. This proves that our method can defend different kinds of transfer schemes.
>
> [r1] ICLR’21 - "Undistillable: Making A Nasty Teacher That CANNOT teach students."
>
> [r2] “Reducing the Teacher-Student Gap via Spherical Knowledge Distillation.”

---

> > ### Comment · Reviewer_w6uB · 2021-08-23
> > **Response to rebuttal**
> >
> > Thanks for the detailed response. All my concerns have been addressed.

---

### Official Review · Reviewer_xU5M · 2021-07-16

**Rating:** 7
**Confidence:** 2

**Summary:**

Three different approaches to embed ownership information in the topological structure of sparse neural networks (i.e. lottery tickets) and their resilience against specific attacks are presented.

**Limitations And Societal Impact:**

Yes.

**Main Review:**

Strengths:
+ Ownership verification of lottery tickets is a novel and relevant problem.
+ Three proposals to embed ownership information into topological structure are presented.
+ Their resilience against different types of attacks is tested.
+ Experiments on CIFAR-10 and CIFAR-100 showcase the applicability of the proposals.

Weaknesses:
- Ownership verification is not a topic that the average NeurIPS community members are familiar with. A more extended introduction that explains the basic concepts in more detail would be helpful. Especially the concept behind trigger sets could be better explained.
- All methods assume access to the training and test data. Often the real issue is sharing privacy sensitive data or claiming ownership of the data, because this data is the key to developing powerful ML models.
- In the problem set-up: Is assumed that the training of neural networks easily feasible or not? This is a relevant question to judge the utility of the first proposal (with key and lock masks).
- On the one hand, already the knowledge of one of the masks could ease the training involved in finding a full lottery ticket because only the missing entries would need to be identified. This does not seem to be a very strong protection of the LT knowledge.
- On the other hand, schemes that need retraining for verification seem to be computationally quite expensive.
- The second proposal limits the minimum sparsity of a lottery ticket.
- Couldn't another subnetwork of the lottery ticket also be confused for a QR code? Even a random one?
Alternatively, what is the strategy if no subnetwork of the LT is similar enough to the encoded QR code?
This proposal seems to be limited by the actual topological nature of the LT.

Points of minor critique:
- Some of the ideas have been proposed for deep neural networks in similar ways.
- A couple of definitions are not precise and the s.
- The machine learning community has put a lot of effort into creating a culture of freely sharing results and trained models.
In this context, ownership verification seems not as obviously important as the authors claim on p. 3.

Post rebuttal:
I acknowledge that I have read the reviews + authors' responses and have raised my score accordingly.

**Time Spent Reviewing:**

3 hours

---

> ### Author Response · Authors · 2021-08-10
> **Response to Reviewer xU5M**
>
> We thank this reviewer for appreciating our novelty in solving this model ownership verification problem. Below is our detailed response:
>
> **[Cons 1:Detailed Explanation on Background]**
>
> Thanks for the valuable suggestions. We will update with a more detailed version of those basic concepts and the logic behind those watermarking methods.
>
> **[Cons 2: Privacy of the data]**
>
> Thank you for the point! We agree that data privacy is important; meanwhile, even the training data is publicly available, training modern ML models on those data can be extremely costly and associated with very high economical and energy resource costs, e.g. think of the training of large language models like BERT and GPT-3, or the state-of-the-art models on video or multi-modality datasets. Not to mention the efforts needed to carefully implement those training and tune hyperparameters in order to achieve the best results. Hence any well-trained model on those datasets can become a valuable asset and a significant investment of the owner institution. That makes ML ownership verification remain a necessary research topic, even the involved data is public. [r1,r2]
>
> Then, since finding winning tickets requires multiple rounds of the train-prune-retrain process, it is extremely costly to find lottery tickets, even more than training the dense model itself. The even higher investment for finding lottery tickets necessitates their protection. We note that recently a number of efforts have been devoted to finding lottery tickets from the above-mentioned big models like BERT, e.g., [r3].
>
> Your suggestion on considering private training data unavailable to the attackers is very interesting to us and will make our next-step research topic. Methodologically, we plan to take references from similar problem domains such as data-free knowledge distillation.
>
>
> **[Cons 3: Feasibility of Scheme 1]**
>
> It is true that currently, we assume that training neural networks is feasible for Scheme 1 since we are distributing the network’s initialization and its sparse mask in our setting. However, our methods&schemes are generalizable and can be adapted to settings where training is infeasible, such as extremely large models or training data that are unreachable. Specifically, under the infeasible training assumption, we will distribute the trained network’s weight and its sparse mask. To further convince the reviewer, we conduct experiments that apply Scheme 1 to the trained model. On CIFAR-10 with ResNet-20, the performance will drop 0.18% when we use a key mask with a relative sparsity of 5%, and 0.77% when the relative sparsity is 10%. The results suggest that our Scheme 1 is also capable of handling the setting of protecting trained models.
>
> **[Cons 4: Can the mask be recovered easily?]**
>
> Although some positions of remaining weights (i.e., “1” elements in the binary mask) on the sparse masks are shown to attackers, it is still infeasible for them to easily find out what the true masks are because:
> - The number of possible 0-1 combinations (i.e., “0”/“1” indicate the position of pruned/remaining weights) is enormous, so it is impossible to enumerate all the possible masks to find out the true one;
> - Given that the brute-force method is infeasible, the attackers can only perform the same tedious iterative magnitude pruning method again to find the true mask. Although some elements of the masks are publicly shown, repeating the train-prune-retrain process multiple times is inevitable. Therefore, the attacking cost is close to finding a new extreme winning ticket, which will lower the practical benefit of attacking.
>
> **[Cons 5: Computational cost of Schemes]**
>
> Scheme 3 is a black-box setting that needs no access to the weights, so we can verify the ownership with only inference on the trigger set. Scheme 2 also only needs little computation overhead to extract the information embedded in the sparse structure.
>
> Using Scheme 1 for verification does need to train the network, but it is like an organic “byproduct” for users to normally use the lottery ticket - they would need to train the network before properly using it anyway. Once the users have trained the model under Scheme 1, they can immediately know the validation of ownership from the performing model. If the performance is lower than the given performance of the extremely sparse winning ticket, the validation process is considered to be failed. In short, there is no extra overhead for lottery ticket users to validate it through Scheme 1.
>
> **[Cons 6: Limitation of minimal sparsity? No similar subnetworks for building QR codes?]**
>
> As our work is the initial step of innovation on this topic, many open problems are supposed to be raised to further our problem setting, application scenarios, and performance levels. Your question is a good one that we will explore in immediate future work.
>
> Our preliminary idea is to split the structural signature into multiple layers or develop more sophisticated algorithms to scatter information. For example, we can split the code into pieces and scatter them in the corners of each convolutional layer, which is easier to tune the mask.
>
> Moreover, we have conducted experiments on VGG-16 and shown that Scheme 2 can be successfully applied to sparse networks over 90% sparsity. Specifically, the sparsity of the extreme winning tickets for VGG-16 on Tiny ImageNet is 93.19%, and applying our Scheme 2 will only cost a negligible performance drop (<0.2% accuracy). We think a sparsity over 90% seems to cover already for most achievable sparsity levels in practice.
>
> **[Minors 1: Borrowing previous work?]**
>
> We admit that some concepts, such as the trigger set and the notion of the verification process, already existed in literature. But we also want to share that embedding QR codes in network architectures for model ownership verification is a brand new idea in this field.
>
> Note that our proposal mainly utilizes the structural information of the pruned sparse network, which “in contrast most of the existing works are based on watermarking either the weights of the model or watermarking labels of a trigger set. “ (refer Reviewer s6wM).
>
> **[Minors 2: Model ownership verification versus open souring culture]**
>
> The authors totally agree on the value of open-source sharing in the ML community, and they are themselves active open-source contributors. Meanwhile, we believe the necessity of studying model ownership is to give people the “freedom of choice”: to publicly share or to retain IP, based on their different goals and contexts. We firmly believe the former is the healthiest way for the academic; while the latter might be necessary for companies to build their commercially competitive edges, and can stimulate their further investment into ML.
>
> **[Minors 3: Definitions are not precise]**
>
> Thanks for pointing this out! We will certainly improve our writing quality and the definition of terms in the final version.
>
> [r1] NeurIPS’19 - “Rethinking deep neural network ownership verification: Embedding passports to defeat ambiguity attacks.”
>
> [r2] ICMR’19 - “DeepMarks: A Secure Fingerprinting Framework for Digital Rights Management of Deep Learning Models.”
>
> [r3] NeurIPS’20 - "The lottery ticket hypothesis for pre-trained bert networks"

---

> > ### Comment · Reviewer_xU5M · 2021-08-17
> > **Response to rebuttal**
> >
> > I thank the authors for their detailed answer and therefore raise my score.

---

### Official Review · Reviewer_YzBW · 2021-07-22

**Rating:** 6
**Confidence:** 2

**Summary:**

Paper proposes methods to protect the winning tickets of the LTH

**Main Review:**

Paper proposes methods to protect the winning tickets of the lottery ticket hypothesis. Being not an expert in the area (model IP protection and verification), my evaluation is based on my shallow understanding of the work. Paper is clearly written, and is easy enough to understand.

Results seems to show good performance after embedding where one doesnt loose more than 0.3% of the baseline accuracy (for scheme V2). There are a few grammatical and structural mistakes,

- On line 293, "Fig 2 show that results of directly re-train the model"
-  Table 4, there is no percentage inside brackets

From a philosophical point of view, I wonder if there can be discussions around the pros/cons of protecting such models, i.e. what does one gain by stealing the ticket?

**Time Spent Reviewing:**

2

---

> ### Author Response · Authors · 2021-08-10
> **Response to Reviewer YzBW**
>
> We thank Reviewer YzBW for appreciating our contribution. We promise to update the grammatical errors in the final version of our manuscript.
>
> From a philosophical point of view, we believe it is necessary to develop model ownership verification methods, since finding winning tickets requires multiple rounds of the train-prune-retrain process, it is extremely costly to find lottery tickets [r1,r2], even more than training the dense model itself. Protecting those high-quality sparse networks can protect the individuals’/companies’ interests who invested in finding them.  Since our approach was shown to cause minimal-to-no degradation to the model performance, we believe this is highly desirable to add protection to the found lottery tickets. Our motivation and novelty of lottery ticket protection is highly appreciated by all other reviewers.
>
> We hope you could kindly check our response and other peer reviews, and we would highly appreciate it if you could then give our work a more positive re-assessment. Thank you!
>
> [r1] ACL’19 - “Energy and Policy Considerations for Deep Learning in NLP”
>
> [r2] ICLR’21 - “Pruning Neural Networks at Initialization: Why are We Missing the Mark?”

---

> > ### Comment · Reviewer_YzBW · 2021-08-16
> > **Thanks**
> >
> > Thanks for the response. I have increased my score accordingly.

---

### Official Review · Reviewer_xVRk · 2021-07-26

**Rating:** 7
**Confidence:** 3

**Summary:**

Winning tickets can often be costly to discover, so this paper explores the issue of copyright protection for winning tickets -- protecting against IP infringement on the owners of a winning ticket. Namely, this paper studies “lottery verification” by analyzing sparse topology of a winning lottery ticket network with the use of several graph-based signatures (i.e., verifiable patterns that can be embedded into the weights/predictions of the winning ticket). Further, these attacks are robust to fine-tuning and extra pruning, which is extremely important for robust validation of winning tickets.While this area has been extensively studied for neural networks in general, this paper seems to be the first to study ownership verification for winning tickets/sparse neural networks. They propose three separate methods for ownership verification: key masks, embedded signatures, and trigger-based methods. Each of these methods are evaluated empirically on the CIFAR datasets using ResNet model variants.

**Limitations And Societal Impact:**

The authors address limitations and there is no negative societal impact.

**Main Review:**

I begin my review by emphasizing that I am completely open to author feedback upon my comments. My final score will be mostly based upon discussion with authors regarding my points outlined below.

General Opinion:
The paper has very good motivation and an interesting topic. But, I think it needs extra work before being accepted. The methods 2 and 3 of ownership verification (i.e., embedding signatures and adding triggers to the network) seem to lack novelty -- these approaches (from what I understand) are taken directly from work on ownership verification for neural networks in general and applied to sparse networks. However, the 1st method of ownership verification that decomposes the mask into separate groups seems to be novel and empirical results are solid. If possible, I recommend the following: (1) emphasize the novel ownership verification components more (i.e., method 1) and extend experiments in this area (2) extend experiments to consider more than one architecture/dataset (i.e., right now all experiments are CIFAR+ResNet) (3) consider different pruning approaches (e.g., structured, channel-based pruning). I believe with extra work this paper will be a good contribution to the community.

Pros:
- The study of ownership verification based on the sparse structure/tology of a winning ticket seems to be novel.
- Numerous possible approaches to verification seem to be explored (e.g., OMP and EWP for sparse mask verification, three different schemes outlined in Sec. 3.4). The most novel approach (i.e., that is not a direct extension from previous work) seems to be the V1 method, which specifically studies locking pruning masks from use without permission.
- Embedding the signature within the low-level convolutional layers is an interesting/clever idea.
- Analyzing the robustness to fine-tuning is very relevant/important, and I find this analysis very useful. The ablation section is well-done in general.

Cons:
- The writing of the manuscript seems like it generally needs to be improved.
- Experiments seem somewhat limited -- only CIFAR+ResNet. But, different depths of ResNets are considered.
- For the V3 method, the performance drop on CIFAR100 seems somewhat significant (i.e., 1% performance decrease). Possibly this can be eliminated with some better tuning.
- V2 and V3 seem to be direct extensions from previous work if I am not mistaken.


Questions:
- The definition of an “extremely sparse ticket” is a bit confusing, and I have never seen this anywhere. Was this just a definition created by authors? Or was this inspired by previous/similar definitions?
- Lines 172-173: In regard to these two ways of protecting winning tickets, it seems that the first method (i.e., method a) is invalid. This is because a “mask” for a winning ticket is useless without the associated weights initialization. This is shown throughout LTH literature that re-initializing weights and training the sparse network will not work well. So, why is there a need for two cases here and what is the difference between them?
- The pruning process that is used is somewhat unclear. Is this just unstructured weight pruning?


Minor Comments:
- There are grammatical errors throughout the paper that the authors should fix within the final version (e.g., lines 5, 7-8,  91-93, etc.).
- Related work never mentions structured vs. unstructured pruning. It only mentions pruning individual weights based on their importance.
- Lines 202-203: it would be nice to cite some work that shows the cost of training certain models here (e.g., https://arxiv.org/abs/2004.08900?source=techstories.org, or other work that references carbon footprints etc.)

**Time Spent Reviewing:**

2

---

> ### Author Response · Authors · 2021-08-10
> **Response to Reviewer xVRk**
>
> Many thank Reviewer xVRk for appreciating our motivation, and our Scheme 1 for novelty. Here are our detailed pointwise responses to your questions:
>
> **[Cons 1. The novelty of Schemes. ]**
>
> Thank you for acknowledging the novelty of our proposed Scheme 1. As you suggested, we have conducted extra experiments on Scheme 1 on new datasets and new architectures, including ResNet-50 on ImageNet, VGG-16 on CIFAR-10, and VGG-16 on Tiny ImageNet. Consistent observations can be drawn from the new extended results (refer to [Cons 2] for more details).
>
> For Scheme 2 and the main part of Scheme 3, we argue that it is not a direct extension of the existing works. Embedding QR codes (i.e., structural signatures rather than previous weight signatures) in neural network architectures has never been studied before, and we are the first to bring them into the ownership verification problem of deep neural networks. The QR codes design is greatly aligned with the nature of sparse structures, which makes it an original, reasonable, and effective solution to the problem.
>
> Furthermore, although the concept of the trigger set (i.e., a part of Scheme 3) is taken from previous literature, we think it should not affect the overall originality of our paper. The main contribution&novelty of this paper is to show that the extremely sparse winning tickets can claim their ownership by appropriately leveraging structural information for the model verification. Such structure-based verification schemes (i.e., Scheme 2 and the main part of Scheme 3) are also highly acknowledged by reviewer s6wM as “this paper gives novel methods for ownership verification using the structural information of the pruned network”.
>
> **[Cons 2. More results of extra architecture (VGG) and dataset (ImageNet)]**
>
> We have conducted multiple experiments with ResNet-50 on ImageNet (larger datasets), and with VGG-16 (various architectures) on CIFAR-10 and Tiny ImageNet (larger datasets). The performance of extreme winning tickets and their sparsity are presented in the below tables.  From the results we can see that our schemes can work very well with extra architectures and datasets, showing a great generality. Specifically, the performances of trained models without key masks have dropped considerably (over 3%) with a relative sparsity of less than 8%. Scheme 2 and Scheme 3 can also show the great ability of protection on VGG-16, indicating the effectiveness of our framework.
>
>
> | Architecture + Dataset | Baseline | Test Accuracy | Sparsity |
> | :-: | :-: | :-: | :-:|
> |ResNet-50 + ImageNet | 75.96% | 75.97% | 83.03% |
> | VGG-16 + CIFAR-10 | 93.06% | 93.10% | 98.56% |
> | VGG-16 + Tiny ImageNet | 57.93% | 58.12% | 93.19% |
>
>
> ResNet-50 + ImageNet with Scheme 1 (OMP):
>
> | Relative Sparsity | Accuracy | Add-back Accuracy |
> | :-: | :-: | :-: |
> | 0% | 75.97% | 75.97%|
> | 0.4% | 75.39% | 75.32%|
> | 5% | 72.88% | 74.57%|
>
>
> VGG + CIFAR-10 + Scheme 1 (OMP):
>
> | Relative Sparsity | Accuracy| Add-back Accuracy |
> | :-: | :-: | :-: |
> | 0% | 93.10% | 93.10% |
> | 8% | 88.63% | 91.96% |
>
>
> VGG + Tiny ImageNet + Scheme 1 (OMP):
>
> | Relative Sparsity | Accuracy| Add-back Accuracy |
> | :-: | :-: | :-: |
> | 0% | 58.12% | 58.12% |
> | 8% | 48.97% | 52.86% |
>
> VGG + Tiny ImageNet + Scheme 2:
>
> | Pruning Ratio | Accuracy| Decode-able? |
> | :-: | :-: | :-: |
> | 0% | 57.95% | Yes |
> | 5% | 57.82% | Yes |
> | 10% | 57.26% | Partly |
>
> VGG + Tiny ImageNet + Scheme 3:
>
> | Pruning Ratio | Accuracy | Trigger-set Accuracy |
> | :-: | :-: | :-: |
> | 0% | 57.93% | 100% |
> | 5% | 57.83% | 100% |
> | 10% | 57.56% | 100% |
> | 20% | 56.44% | 100% |
> | 50% | 8.24% | 31% |
>
> **[Cons 3. More pruning methods?]**
>
> The reason for adopting iterative magnitude pruning (IMP) in this paper is mainly following the standard of almost all LTH works [r2,r3]. Note that there is no current work capable of using structure pruning to find winning tickets. Thus, this is a very challenging open question beyond the scope of this paper.
>
> To further convince the reviewer, we conduct post-training pruning (not the lottery ticket hypothesis) experiments that iteratively using saliency-based pruning methods defined in [r1] on ResNet-20 on CIFAR-10, as well as an experiment with network slimming method on VGG-16 on CIFAR-10. After we find sparse masks, we apply Scheme 1 to the masks to see how our Scheme 1 works (note that sparse subnetworks will be rewinded to their same initialization).
>
> The sparse network we obtained has a sparsity of 36% and a test accuracy of 90.91%. With Scheme 1 (OMP), the performance of this sparse network drops 0.14% when the relative sparsity of the key mask is 3% and 0.61% with a relative sparsity of 6%. For the network slimming experiment on VGG-16, we first obtain a slimmed network that has 50% channel-pruned and the performance is 87.61%. When we keep out 5% channels as key masks, the performance is dropped to 86.88%. When we keep out 10% of channels as key masks, the performance is dropped to 86.56%. This shows that our protection schemes can generalize to other sparse masks, which is an extra bonus of our proposal.
>
> **[Cons 4. Paper writing]**
>
> Thanks for your kind reminder. We promise to do more proofreading and improve the quality of writing in our revision.
>
> **[Cons 5. Tuning of Scheme 3]**
>
> Thanks for your suggestion and we agree with you that this gap can be eliminated with some better tuning. We tried some tuning (e.g., when to inject the trigger images, the ratio of the trigger images in the batch, etc.) during the rebuttal period, and now the performance drop can be reduced to 0.73%. We will keep improving the training protocols to reduce the performance drop further.
>
> **[Cons 6. Definition of extreme winning tickets]**
>
> The concept of “extremely sparse winning tickets” is greatly inspired by [r4], but in our work, we proposed a more rigorous definition. The “extremely sparse winning tickets” is the winning ticket that cannot be pruned further otherwise the performance will become lower than the dense counterpart. Please refer to Line 128-131 for a more detailed definition. We will further clarify it in the final version.
>
> **[Cons 7. Difference between two protecting classes]**
>
> We want to assure you that our protecting class 1 is valid since we also distribute the locked version of initialization weights to users (refer to Line 242). You are correct about the relationship between masks and initialization - the LTH sparse mask cannot work well without the same initialization. This is also why we distribute the mask together with the corresponding initialization.
>
> The first scenario is needed because when the models are training feasible, users can train sparse models by themselves for different purposes. For example, distributed masks can also be utilized in pretrain-and-transfer settings as demonstrated in [r3], which first find tickets on a pre-trained task and then fine-tune it on diverse downstream tasks.
>
> The difference between the two scenarios lies in the distributed weights. For the first one, only the sparse mask and part of the initialization weights (refer to Line 242) are given. While in the second scenario, the **trained** sparse networks are dispatched, since some massive models are infeasible to train by users themselves, such as GPT-3 and BERT due to a large number of parameters or unreachable training data.
>
> **[Cons 8. Pruning definition]**
>
> Yes, we used the standard iterative magnitude pruning commonly used in LTH literature [r2,r3]. We promise to add a clearer description of the pruning process in our final version.
>
> **[Cons 9. Other minor comments]**
>
> We truly thank the reviewer for pointing out our grammatical errors and the suggestions on related works and references. We will update our manuscript with more detailed related work and more references on environmental issues in the final version of our manuscript.
>
> [r1] CVPR’19 - "Importance estimation for neural network pruning".
>
> [r2] ICML’18 - "The Lottery Ticket Hypothesis: Finding Sparse, Trainable Neural Networks."
>
> [r3] NeurIPS’20 - "The lottery ticket hypothesis for pre-trained bert networks"
>
> [r4] ICLR’ 21 - "Gans can play lottery tickets too."

---

> > ### Comment · Reviewer_xVRk · 2021-08-17
> > **Response to rebuttal**
> >
> > I believe the authors did a great job of addressing my major concerns. As mentioned in my review, the scope/motivation of the work is very valuable, and I am more confident in this work given the new results and explanations. After viewing the rebuttal/discussion, other author reviews, and other rebuttals, I have decided to increase my score accordingly. Thank you very much to all authors for the useful clarifications.

---

### Decision · Program_Chairs · 2021-09-27

**Decision:**

Accept (Poster)

**Comment:**

We thank the authors for this submission. Overall, the paper is about a topology-based ownership verification mechanism that can prevent lottery-ticket theft under various verification schemes and attacks.

The paper well-motivates the approach. The authors have provided extensive responses to the concerns raised and the AC + reviewers really thank them for their effort. Overall, the new results obtained during the rebuttal definitely improve the quality of the paper. We all believe that the inclusion of these results during the rebuttal period is something that does not heavily change the message of this paper.

There was discussion and consensus that this work is interesting. Having in mind issues/concerns raised by the reviewers, the main points of reviewers during further discussion were that this paper deserves publication, given the promised fixes by the authors during the discussion period.